# A natural constant predicts survival to maximum age

Manuel Dureuil [1,2✉] & Rainer Froese [3]

Information about the survival of species is important in many ecological applications. Yet, the estimation of a species' natural mortality rate $M$ remains a major problem in the management and conservation of wild populations, often circumvented by applying empirical equations that relate mortality to other traits that are more easily observed. We show that mean adult $M$ can be approximated from the general law of decay if the average maximum age reached by individuals in a cohort is known. This is possible because the proportion $P$ of individuals surviving to the average maximum age in a cohort is surprisingly similar across a wide range of examined species at 1.5%. The likely reason for the narrow range of $P$ is a universal increase in the rate of mortality near the end of life, providing strong evidence that the evolutionary theories of ageing are the norm in natural populations.

[1] Department of Biology, Dalhousie University, 1355 Oxford St, Halifax B3H 4R2, Canada. [2] Sharks of the Atlantic Research and Conservation Centre, 279 Portland Street, Dartmouth B2Y 1K2, Canada. [3] GEOMAR Helmholtz Centre for Ocean Research, Düsternbrooker Weg 20, Kiel 24105, Germany. ✉email: ManuelDureuil@gmail.com

Exponential decay, like its opposite exponential growth, has been shown in a wide variety of situations, such as the decay of radioactive matter[1], the decrease of atmospheric pressure with height[2], the decline in number of test tubes due to random breakage[3] and even the disappearance of beer froth[4]. Here, we show that the same exponential law of decay can also be used to estimate the average rate of adult natural mortality in wild populations of various species.

The exponential law of decay is often given as percent decline per period of time:

$$N_t = N_{t0}(1 - r_p)^t, \quad (1)$$

where $N_t$ is the number of items of interest $N$ after time $t$, $N_{t0}$ is the number of items at time $t_0$, and $r_p$ is the rate of decline in percent, expressed as decimal. In many scientific applications, $r_p$ is replaced by the instantaneous rate of decline $Z$, which has the useful property that several different rates can be combined by addition, e.g., $Z = Z_1 + Z_2 + Z_3$, resulting in the simplified equation:

$$N_t = N_{t0}e^{-Z*(t-t_0)}, \quad (2)$$

where the number $e$, often referred to as Euler's number, is a mathematical constant. In populations of living beings, $Z$ is the rate of mortality at any point in time, e.g. with the time unit year$^{-1}$. $Z$ can be determined from Eq. (3) if $N_t$ and $t$ are known:

$$Z = -\frac{log_e(N_t/N_{t0})}{(t - t_0)}, \quad (3)$$

with $log_e$ being the natural logarithm to the base $e$. Note that in the form of discrete instead of continuous time intervals the survival probability $S$, or per capita death rate $D$, which is also referred to as percentage mortality, is used. These are related to $Z$ via $S = e^{-Z}$ and $D = 1 - e^{-Z}$. The total mortality $Z$ is often subdivided such that $Z = M + F$, where $M$ represents all natural causes of death and $F$ represents mortality caused by humans e.g. through hunting or fishing. The rate of natural mortality $M$ is of key importance in population dynamics and the sustainable management of natural resources[5–8]. Yet, the natural mortality or natural survival rate is exceedingly difficult to estimate in wild populations, often requiring data-intensive and costly approaches[5,9]. Natural populations can also display different types of survivorship curves, from linear with about constant mortality throughout life, to U-shaped with increased mortality early and late in life[10]. In many practical applications a common simplification to aid the estimation process is the assumption of a constant mortality rate[6,11–15], representing the mean mortality rate across all age classes and, given that juvenile and senescence phases are typically short, the approximate mortality rate of adults during the main reproductive phase.

Despite the central role of natural mortality in estimating extinction risk or managing wild populations, reliable estimates of $M$ are missing for the vast majority of species[16]. Hence, a more general approach accounting for data-limitations when estimating the average natural mortality rate in wild populations is required. We propose to estimate the average natural mortality rate during the main reproductive phase from the observed maximum age, $t_{max}$, taken ideally as the mean across maximum ages observed over a number of years or the mean age of the most long-lived 10% in a cohort[17], i.e., within a group of individuals all born in the same year, and here approximated for natural populations by the mean age of the eldest individuals reported for a species. When replacing $t$ with $t_{max}$ in Eq. (3), and in the absence of significant anthropogenic mortality so that $M \approx Z$, the missing piece of information is the proportion $P = N_{tmax}/N_{t0}$ of individuals surviving to the average maximum age:

$$M = -\frac{log_e(P)}{t_{max}}. \quad (4)$$

Note that in ecological applications the proportion of organisms surviving from birth to age $x$ might be labelled $l_x$, and $M$ or $Z$ referred to as hazard rates or risks. From Eq. (4) it follows that if $t_{max}$ and $M$ are known $P$ can be calculated from $P = e^{(-M*t_{max})}$.

It has been proposed for various species that the average maximum age is reached by about $1-5\%$ of individuals in a cohort, including fish[18–22], reptiles[23], birds[24] and trees[25]. Indeed, if the proportion of individuals surviving from birth to the average maximum age in a cohort, $P$, would be the same for different species within different taxonomic groups, and if mortality follows an exponential decay, then knowledge of $P$ would make Eq. (4) a direct and universal natural mortality estimator, requiring only observations on maximum age. Hence, the purpose of this study is to estimate $P$ from wild and natural, not hunted or harvested, populations for which $t_{max}$ and $M$ are known and to draw conclusions about its variability and its general applicability.

## Results and discussion

We found a surprisingly narrow range for the proportion of individuals surviving from birth to the average maximum age in a cohort, $P$, across 202 species from 6 vertebrate classes (Supplementary Table S1; Fig. 1), with a median $P$ at 0.015 and 90% of the estimates falling between 0.0004 and 0.11. The estimates of $P$ were not significantly different across all classes (Kruskal–Wallis rank sum test $p$-value = 0.6), suggesting that 1.5% of individuals belonging to the same cohort survive to the average maximum age of that cohort, independent of vertebrate taxonomy. Moreover, there are various examples from other species including algae, plants, insects, marine invertebrates and humans where the proportion surviving to maximum age is also equal or close to 0.015 (Table 1).

Although the data for *Homo sapiens* and non-vertebrate species in Table 1 are not representative, these examples show that the $P$ estimate of 0.015 established from comprehensive vertebrate data can also be found in other groups, indicating that there is a common mechanism at work that determines the probability of survival to average maximum age in a cohort, despite vastly different life histories and different strategies of survival.

With a constant mortality rate the oldest observation should be a log-function of the initial cohort or sample size[26,27] (see straight declining lines in Fig. 2), and with only the one eldest individual surviving in a cohort, $N_{tmax}$ becomes 1 and therefore $P$ becomes a function of the initial numbers, $P = 1/N_{t0}$. Instead, for the average maximum age in a cohort, the proportion of survivors becomes largely independent of initial numbers at 1.5%, as indicated in Fig. 2, and as suggested by the variety of species examined in the present study, which are reflective of a variety of different starting numbers of individuals at birth. The most convincing explanation for this observation is an exponential increase in mortality rate as maximum age is approached (see dotted convex curves in Fig. 2), given that otherwise the age of the eldest individual in a cohort would far exceed the average maximum age (see dotted straight declining lines in Fig. 2). Such an increase in mortality is also predicted by evolutionary theory, because natural selection against intrinsic causes of mortality becomes weaker as the contribution to future generations diminishes with increasing age[3,28–31]. Here we provide further evidence across a wide range of different species, that increasing mortality towards maximum age, called actuarial senescence, is likely a common phenomenon in natural populations.

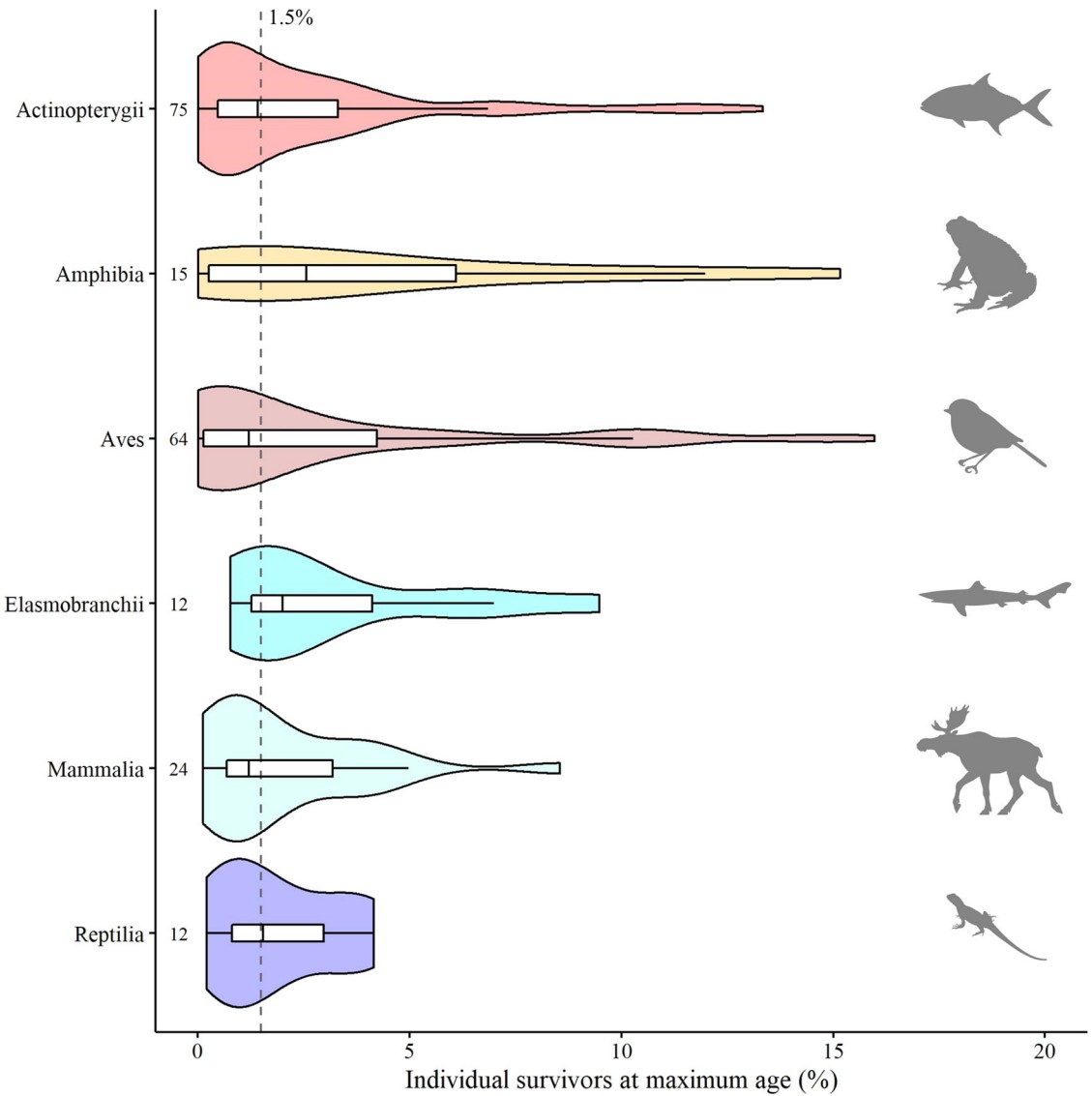

**Fig. 1 Survival to maximum age in natural populations of vertebrates.** The median survival to average maximum age in a cohort is shown per vertebrate class as a vertical solid line within each boxplot, the 25th and 75th percentiles are the lower and upper bound of the boxes, respectively. Violin plots indicate the density distribution of the data points, the numbers refer to the sample size (number of species examined) per vertebrate class. The dashed vertical line indicates the median survivorship to average maximum age in a cohort across all vertebrate classes, at 1.5% survivors. Animal silhouettes represent the different vertebrate classes and are available from PhyloPic (http://www.phylopic.org), public domain.

To further illustrate this, the hypothetical population in Fig. 2 was modelled with life history data of Atlantic herring (*Clupea harengus*)[32]. If there were no increase in mortality late in life, the age of the oldest individuals in the cohort would be a direct function of cohort size[26,27] (see Equation S1). In Atlantic herring, where the numbers of individuals joining the adult population (recruits) are in the thousands of millions[33], it would follow that the oldest individual would need to be far older than 100 years (Fig. 2). Moreover, in the case of negative senescence Atlantic herring would approach an indefinite life span (Fig. 2). If on the other hand mortality increases towards maximum age, one obtains more reasonable predictions for $t_{max}$ (Fig. 2), closer to the observed maximum ages for Atlantic herring of up to 26 years of age[32].

The mortality budget, where on average 98.5% of the cohort die before reaching maximum age, also suggest that species with additional mortality early in life could be more resistant to develop actuarial senescence before the average maximum age in a cohort. If high mortality rates early in life would use up much of

the 98.5% mortality budget, then additional mortality imposed later in life could result in a survivorship at average maximum age below 1.5% (Fig. 2). Accordingly, if mortality is only added later in life, then the development of actuarial senescence before the average maximum age in a cohort could be favoured, so that the numbers of individuals surviving to average maximum age is reduced to 1.5% (Fig. 2). This interaction between additional mortality at different stages in life and the tendency to develop senescence has been described previously[34]. Independent from the mortality trajectories before 1.5% survivorship, the predicted theoretical maximum age that can be reached by 1 individual (oldest observation) would become more realistic when mortality increases very sharply at 1.5% survivorship (Fig. 2).

There is considerable evidence of senescence in wild mammals, birds, reptiles and fish[35–40], and limited evidence in wild amphibians[38,39]. Evidence for actuarial senescence is also emerging for wild insects[38,41,42], and higher levels of mortality in older individuals are common in plants[43]. Nevertheless, it can be seen by the variability in Fig. 1 and from studies elsewhere[44], that

**Table 1 Survival to maximum age across various species.**

| Group | Species | Common name | P |
|---|---|---|---|
| Alga | *Laminaria digitata* | Oarweed | 0.015 |
| Arachnid | *Latrodectus mactans* | Black widow | 0.016 |
| Bivalve | *Panope abrupta* | Geoduck clam | 0.013 |
| Bivalve | *Panopea abbreviata* | Southern geoduck | 0.010 |
| Bivalve | *Siliqua patula* | Pacific razor clam | 0.013 |
| Bivalve | *Spisula polynyma* | Alaska surf clam | 0.009 |
| Crustacean | *Callinectes sapidus* | Blue crab | 0.014 |
| Gastropod | *Cerithidea decollata* | Truncated mangrove snail | 0.014 |
| Insect | *Aedes aegypti* | Yellow fever mosquito | 0.015 |
| Insect | *Apis mellifera* | European honey bee | 0.013 |
| Plant | *Cyperus rotundus* | Nut grass | 0.010 |
| Plant | *Phlox drummondii* | Drummond's phlox | 0.022 |
| Plant | Various tree species | Trees | 0.020 |
| Human | *Homo sapiens* | Modern human | 0.021 |
| Starfish | *Archaster angulatus* | Sea star | 0.010 |

Median survival to the average maximum age in a vertebrate cohort, $P = 0.015$ (Fig. 1, Table S1) is similar to the survival estimates obtained from literature for various other species shown in this table, where the median $P$ equals 0.014. Details and references are provided in the Supplementary Information (Table S2).

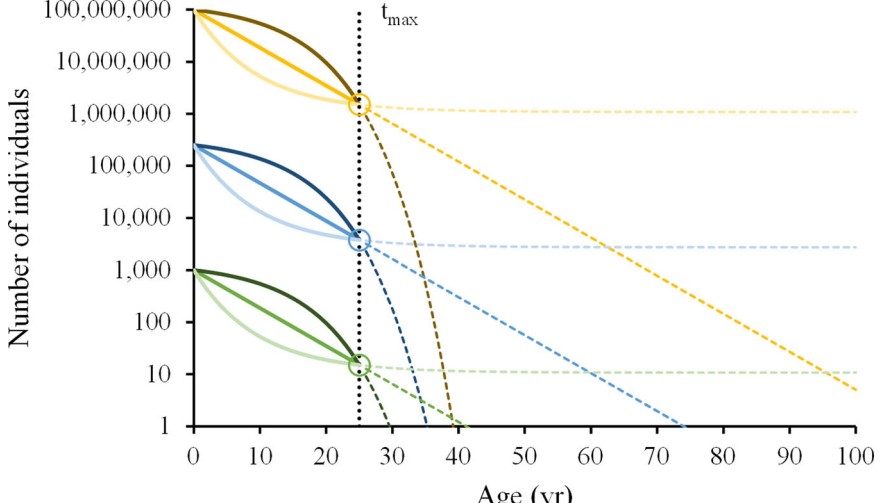

**Fig. 2 The number of survivors to a certain age under different survivorship patterns and initial cohort (or sample) sizes.** Individual survivors to the average maximum age in a cohort, $t_{max}$, of 25 years (black dotted vertical line), representing 1.5% of the initial numbers ($P = 0.015$, open circles). Shown are scenarios for constant mortality, mortality increasing with age (dark colours) and mortality decreasing with age (light colours). The coloured dotted lines show the expected trajectory of these scenarios from 1.5% survivorship until the theoretical maximum age reached by 1 individual.

other patterns may occur in nature. Possible exceptions to the concept of actuarial senescence include a eusocial vertebrate, the naked mole rate[45], some eusocial insects[46], semelparous species such as cephalopods which die immediately after reproduction at an age reached by approximately 50% of the population[47], and some plant species due to unique aspects of their biology[43,48]. Senescence in nature can also be masked by hunting which can create the wrong impression of no or even negative senescence, e.g. if old individuals are disproportionally targeted, and thus additional anthropogenic mortality might change the typical form of the survival curve[49]. However, the present study suggests that most natural populations of wild species show an increase in mortality towards the end of life.

In conclusion, this study provides strong evidence for a surprisingly constant proportion of around 1.5% of individuals surviving to the average maximum age in a cohort, across a wide range of species, from plants and invertebrates to vertebrates. The observed constancy in $P$ and the surrealistically high ages when

mortality remains constant towards the end of life, suggest an increase in mortality towards maximum age (actuarial senescence) as a main driver. The suggested increase in mortality has been regularly observed in experiments and described with the Gompertz[50] or the Gompertz-Makeham[51] model[17,35,41,52–55] and is providing further empirical support that evolutionary theories of ageing[3,28–31] generally apply to natural populations. In practical implementation, $P = 0.015$ appears to be a good approximation to estimate an important parameter in natural resource management and ecology from maximum age information, the average natural mortality rate $M$ of adults during the main reproductive phase. Unlike the often U-shaped mortality curve throughout life, the natural mortality rate during the main reproductive phase can generally be described by the constant average rate with $M = -\frac{\log_e(P)}{t_{max}}$. Note that this equation is derived from the first principle exponential law of decay, with the two variables representing biological traits, i.e. the average maximum age in a cohort $t_{max}$ and the typical proportion $P$ surviving to $t_{max}$.

Interestingly and important for practical applications, the effect of an error in $P$ on the natural mortality estimate is rather small (Fig. S1). i.e., the error in predicted $M$ will be mostly determined by the error in observed $t_{max}$. In the absence of detailed information on the average maximum age in a cohort, the eldest individual found, aged, and recorded might serve as a proxy of $t_{max}$ in wild populations. Given that observations on maximum age are more readily available for many species than estimates of $M$, the presented method is believed to have wide applicability in natural resource management and conservation of species.

## Methods

**Data selection.** A comprehensive database on the average maximum age reached by individuals in a population, $t_{max}$ (years), and the mean adult natural mortality rate, $M$ (year$^{-1}$), across vertebrate classes was built (Supplementary Data 1), mostly by combining existing databases on mammals (Mammalia)[35,36,56,57], birds (Aves)[37], reptiles (Reptilia) (https://datlife.org/), amphibians (Amphibia) (https://datlife.org/; A database of life-history traits of European amphibians: https://www.ncbi.nlm.nih.gov/pmc/articles/PMC4237922/), and marine fish (Actinopterygii and Elasmobranchii)[22]. It is unlikely that in natural populations of wild animals the oldest individual was found, aged, and reported. Therefore, the observed maximum age from the oldest individual found and aged was utilised as a proxy for average maximum age in a cohort of a population or species. Species with high anthropogenic mortality were excluded. For mammals in Promislow[35] and Sibley et al.[36], and for reptiles and amphibians, the original literature composing the data for the adult mortality or survival values was checked and populations that were likely highly impacted by humans, e.g., through hunting or road kills, were excluded. Natural mortality estimates of marine mammals estimated by Ohsumi[56], but considered incorrect in Mizroch[57], were replaced by the estimates in Mizroch[57] or were excluded. For birds, all species with an expected high anthropogenic impact were excluded, as mentioned in Birds of North America, https://birdsna.org. For marine fish, all data came from directly estimated adult natural mortality and maximum age values[22]. Due to limited data for natural populations of reptiles and amphibians in existing databases, observed natural mortality rates and maximum ages were additionally obtained from a literature review. Unlikely cases where only one in 10,000 individuals reached $t_{max}$ or where over 20% of individuals reach $t_{max}$ and died, were excluded (Actinopterygii: $n = 2$; Aves: $n = 16$; Mammalia: $n = 2$). This had no effect on the differences in survival to maximum age $P$ across taxa (Kruskal-Wallis rank sum test $p$-value with all data, $p = 0.7$, and with unlikely cases excluded, $p = 0.6$), and the median $P$ per taxa and the median $P$ over all taxa were identical. In addition, the literature was searched for examples of other species where survival to maximum age was similar to the survival patterns of the examined vertebrate classes. These data were added to investigate wider applicability but are not necessarily representative for their groups.

**Estimation of survival to maximum age.** When several adult natural mortality or survival values were provided in the databases for the same population, the mean was taken. Then, the mean survival or mortality and the mean of the observed maximum ages was taken across sex and populations to obtain one value of survival or mortality and one value of maximum age per species. Next, survival or percentage mortality rates were transformed into annual instantaneous mortality rates (year$^{-1}$) and maximum age was transformed to years, where necessary. Finally, the proportion surviving to average maximum age in a cohort, $P$, was calculated using the observed maximum age and the adult natural mortality estimated per species by rearranging Eq. (4): $P = e^{(-M*t_{max})}$.

**Reporting summary.** Further information on research design is available in the Nature Research Reporting Summary linked to this article.

## Data availability

The data that support the findings of this study are based on literature reviews and therefore publicly available. The calculated proportion of individuals surviving to the average maximum age in a cohort $P$ is provided for each of the examined vertebrate species as electronic supplement.

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

## Acknowledgements
We would like to thank Dr. Boris Worm for discussions and feedback, which helped to improve the manuscript greatly. Research funding was provided by the Ocean Frontier Institute, through an award from the Canada First Research Excellence Fund. R.F. was financed by the German Federal Agency for Nature Conservation (BfN) with funds from the Federal Ministry of the Environment, Nature Conservation and Nuclear Safety (BMU).

## Author contributions
M.D. and R.F. designed the study. M.D. compiled and analysed the data. M.D. and R.F. discussed the results and jointly wrote the manuscript.

## Funding

## Competing interests
The authors declare no competing interests.
