## [Peer Review File · Communications Biology]

Reviewers' comments:

Reviewer #1 (Remarks to the Author):

This is an interesting study, which claims that the proportion (P) of individuals surviving to maximal age (eldest individual found) is similar across examined species at 1.5%, and it is largely independent of initial numbers. To explain this observation, the authors suggest that mortality rates are increasing with age in the wild.

The authors also suggest an equation for estimating the average adult mortality rate in natural populations from the observed maximum age, assuming that the natural mortality rate does not depend on age (exponential decay law model).

There is an obvious logical contradiction between these two mutually exclusive suggestions made by the authors. This apparent contradiction needs to be explained and eliminated.

Minor comment: Because the authors mention humans (*Homo sapiens*) in their study, it is important to indicate explicitly, what is the maximum age for humans (to provide a number and justification, not just references) and their natural mortality rate.

Reviewer #2 (Remarks to the Author):

Title: A natural constant predicts survival to maximum age.

Authors: Manuel Dureuil, Rainer Froese.

The manuscript investigated the proportion of individuals surviving to their maximum age was similar by a wide range of species. Simultaneously, the author pointed out the average natural mortality can be approximated from the law of decay if the maximum age reached by individuals in a population was given. The idea is interesting, but a trivial analysis method is adopted. Authors have to carefully address some points.

Some specific comments:

1. On line 43, "Nt0 is the number of items at time zero t0" Should be changed to "Nt0 is the number of items at time t0". Besides, on line 52, the formula $z = -\log(Nt - Nt0)/(t - t0)$, however, on line 73, the formula 4 changed $z = -\log(P)/t_{max}$, the denominator of formula 3 and 4 is different, please give a detailed and reasonable explanation.
2. There may be some typing errors in the manuscript. For example, "t0" may be corrected as "t0". Please check them carefully. In addition, the format is not standard and the formula should be centered, question of space at the beginning of a paragraph, and so on.
3. On line 56, "The total mortality, Z, is often subdivided in natural Mortality M and mortality caused by humans e.g. through hunting or fishing F, such that $Z = M + F$." Z is divided into two categories. But in an ecosystem, there are competition or predator-prey relationships between animals, and the deaths that result from these two relationships are not negligible. I think these factors should also be taken into account.
4. On line 149, "constant natural mortality tc", I think the explanation of tc is wrong, please give the correct explanation.
5. Please give a detailed derivation of how formula 5 was obtained.

Reviewer #1 (Remarks to the Author):

R1.1: This is an interesting study, which claims that the proportion (P) of individuals surviving to maximal age (eldest individual found) is similar across examined species at 1.5%, and it is largely independent of initial numbers. To explain this observation, the authors suggest that mortality rates are increasing with age in the wild.

The authors also suggest an equation for estimating the average adult mortality rate in natural populations from the observed maximum age, assuming that the natural mortality rate does not depend on age (exponential decay law model).

There is an obvious logical contradiction between these two mutually exclusive suggestions made by the authors. This apparent contradiction needs to be explained and eliminated.

Response R1.1: We thank the reviewer for this positive feedback and for pointing out the apparently circular argument in our suggestions that natural mortality M increases towards the end of life, but that when estimating P we assume constant M and suggest that P derived this way across taxa can also be used in turn to estimate M.

There is indeed a superficial contradiction between estimating a constant mortality rate throughout life and arguing that the surprisingly constant percentage of survivors is caused by increase in intrinsic mortality late in life. In the text, we already stressed that many species have increased mortality rates as juveniles and as adults, often resulting in a U-shaped pattern of mortality-at-age. However, models for management or conservation of wild populations are mainly interested in the mortality of reproducing adults, and mortality during that 'main reproductive phase' is usually sufficiently well described by an average mortality rate. This average mortality rate is approximated by the exponential decay model with t_{max} and P as input. For example, in stock assessments of fish populations, plotting log numbers of survivors over age is nearly always fitted well by a straight line, the slope of which is an estimation of mean total

mortality rate Z , never mind that mortality rates of early life stages are known to be much higher, and mortality rates of oldest individuals are unknown. We have expanded the existing text to explicitly address the apparent contradiction pointed out by the reviewer.

R1.2: Minor comment: Because the authors mention humans (*Homo sapiens*) in their study, it is important to indicate explicitly, what is the maximum age for humans (to provide a number and justification, not just references) and their natural mortality rate.

Response R1.2: We were aware that humans are mostly interested in human mortality rates, but progress in medicine and general safety measures are distorting the numbers, making them not reflective of a natural and “wild” population. Moreover, we wanted to avoid a focus on humans. To illustrate the potentially wide applicability however, we have included one example of a group of humans which seems to follow the general pattern, using the median proportion surviving to the last age group from life tables of Japanese males and females from 1776 to 1795, with age groups being in 5-year intervals. Therefore, one exact number for maximum age was not provided, but details are given in the Supplementary Information.

Reviewer #2 (Remarks to the Author):

Title: A natural constant predicts survival to maximum age.

Authors: Manuel Dureuil, Rainer Froese.

R2.1: The manuscript investigated the proportion of individuals surviving to their maximum age was similar by a wide range of species. Simultaneously, the author pointed out the average natural mortality can be approximated from the law of decay if the maximum age reached by individuals in a population was given. The idea is interesting, but a trivial analysis method is adopted. Authors have to carefully address some points. Some specific comments:

Response R2.1: We thank the reviewer for the constructive critique and respond in detail to all addressed points below.

R2.2: 1. On line 43, "Nt0 is the number of items at time zero t0" Should be changed to "Nt0 is the number of items at time t0". Besides, on line 52, the formula 3 $z = -\log(Nt/Nt0)/(t-t0)$, however, on line 73, the formula 4 changed $z = -\log(P)/t_{max}$, the denominator of formula 3 and 4 is different, please give a detailed and reasonable explanation.

Response R2.2: Changed to "Nt0 is the number of items at time t0". We think the original text was clear about the difference in the denominator, but we are happy to elaborate more on this here: The difference in the denominator is because in formula 4 age t becomes maximum age tmax, whereas t0 becomes 0 (age at birth):

$$z = -\log(Nt/Nt0)/(t-t0);$$

with $t0=0$, $t= t_{max}$, $Nt=N0$, $Nt =Nt_{max}$ this becomes

$$z = -\log(Nt_{max}/N0)/(t_{max}-0);$$

with $P= Nt_{max}/N0$ this becomes

$$z = -\log(P)/t_{max};$$

R2.3: 2. There may be some typing errors in the manuscript. For example, "t0" may be corrected as "t0". Please check them carefully. In addition, the format is not standard and the formula should be centered, question of space at the beginning of a paragraph, and so on.

Response R2.3: We thank the reviewer for pointing this out. We carefully checked the manuscript for typing errors, centered all formulas and made a space at the beginning of each paragraph.

R2.4: 3. On line 56, "The total mortality, Z, is often subdivided in natural Mortality M and mortality caused by humans e.g. through hunting or fishing F, such that $Z = M + F$." Z is divided into two categories. But in an ecosystem, there are competition or predator-prey

relationships between animals, and the deaths that result from these two relationships are not negligible. I think these factors should also be taken into account.

Response R2.4: Competition or predator-prey relationships between animals are included in M. To clarify this we changed the sentence to: "The total mortality, Z, is often subdivided in natural Mortality M representing all natural causes of death and mortality caused by humans e.g. through hunting or fishing F, such that $Z = M + F$."

R2.5: 4. On line 149, "constant natural mortality t_c ", I think the explanation of t_c is wrong, please give the correct explanation. Please give a detailed derivation of how formula 5 was obtained.

Response R2.5: Equation 5 was proposed by Holt (1965) and confirmed by Hoenig (2017), who define t_c as the age above which natural mortality is assumed to remain constant. In populations with high cohort numbers, such as herring, the equation strongly overestimates maximum age and we therefore do not recommend its use. We only included it to demonstrate that assuming no senescence leads to unrealistic results. We have changed the main text and moved the equation to the Supplement section to make this clearer.

Reviewers' comments:

Reviewer #1 (Remarks to the Author):

This is a revised manuscript, which claims that the proportion (P) of individuals surviving to maximal age (eldest individual found) is similar across examined species at 1.5%, and it is largely independent of initial numbers.

I have a naive question: By definition the number of individuals surviving to maximal age (eldest individual found) is equal to 1. Hence the proportion (P) of individuals surviving to maximal age is not a constant, but is a reciprocal function of the initial numbers "N":

$$P = 1/N$$

Please explain whether this argument is wrong, and if so, why.

Reviewer #1 (Remarks to the Author):

R1.1: This is a revised manuscript, which claims that the proportion (P) of individuals surviving to maximal age (eldest individual found) is similar across examined species at 1.5%, and it is largely independent of initial numbers.

I have a naive question: By definition the number of individuals surviving to maximal age (eldest individual found) is equal to 1. Hence the proportion (P) of individuals surviving to maximal age is not a constant, but is a reciprocal function of the initial numbers "N":

$$P = 1/N$$

Please explain whether this argument is wrong, and if so, why.

Response R1.1: We thank the reviewer for raising this point, because it made us realize, that our definitions and explanations in the text were not clear. We reformulated the respective parts in the manuscript, which in our view improved the general understanding of the manuscript greatly.

The definition that the number of individuals surviving to maximal age (eldest individual found) is equal to 1 is correct from an individual (specimen) point of view. However, what has not been made clear enough by us in the text was that we are concerned with a species' average survival to maximum age in a cohort, so the mean/average life span of a species across several individuals, in part from different populations. Each individual has their own maximum age, there is variability among the members of the population. Knowing that on average 1.5% of the initial individuals reach the mean life span in natural populations across species allows us to draw important conclusions about the average adult natural mortality rate in a species or population.

We understand that we needed to elaborate more on this, given that we previously stated that maximum age can be obtained by the eldest individual found and aged, without mentioning that this is used as a proxy, given that we do not have actuaries for wild animals and plants. In the wild, we do not really know the oldest individual, only the oldest found and reported. That blurs the distinction between mean life span and the oldest found. In a pragmatic approach we have either used the oldest individual reported, or a mean of the oldest reported if there were many reports, with maximum age referring to the average maximum life span of a species across different populations and sex. The oldest individual found and reported will not be the 'true' oldest, because the chance of finding that individual is very low, but rather a lower age, which we have used as reasonable proxy for the mean life span of a species.

We believe that this approach to obtain a mean lifespan reflective of natural populations of wild animals is a valid approach given the limited data available.

We have expanded the existing text and rephrased the definition on maximum age to make clear that it is from a species point of view. We hope that this is eliminating the concerns and answering the very valid point raised by the reviewer.